# Optimal Sparsity-Sensitive Bounds for Distributed Mean Estimation

**Zengfeng Huang**
School of Data Science
Fudan University
huangzf@fudan.edu.cn

**Ziyue Huang**
Department of CSE
HKUST
zhuangbq@cse.ust.hk

**Yilei Wang**
Department of CSE
HKUST
ywanggq@cse.ust.hk

**Ke Yi**
Department of CSE
HKUST
yike@cse.ust.hk

## Abstract

We consider the problem of estimating the mean of a set of vectors, which are stored in a distributed system. This is a fundamental task with applications in distributed SGD and many other distributed problems, where communication is a main bottleneck for scaling up computations. We propose a new sparsity-aware algorithm, which improves previous results both theoretically and empirically. The communication cost of our algorithm is characterized by Hoyer's measure of sparseness. Moreover, we prove that the communication cost of our algorithm is information-theoretic optimal up to a constant factor in all sparseness regime. We have also conducted experimental studies, which demonstrate the advantages of our method and confirm our theoretical findings.

## 1 Introduction

Consider a distributed system with $n$ nodes, called clients, each of which holds a $d$-dimensional vector $X_i \in \mathbb{R}^d$. The goal of *distributed mean estimation* (DME) is to estimate the mean of these vectors, i.e., $X := \frac{1}{n} \sum_{i=1}^{n} X_i$, subject to a constraint on the communication cost (i.e. the total number of bits transmitted by all clients).

DME is a fundamental task in distributed machine learning and optimization problems [3, 10, 18, 14, 12]. For example, gradient aggregation in distributed stochastic gradient decent (SGD) is a form of DME. In the standard synchronous implementation, in each round, clients evaluate their local gradients with respect to local mini-batches and communicate them to a central server; the server then computes the mean of all these gradients, which is used to update the model parameters. It is widely observed that the communication cost of gradient exchange has become a significant bottleneck for scaling up distributed training [5, 19, 24]. Therefore, communication-efficient gradient aggregation has received lots of attention recently [1, 2, 15, 23, 26]. DME is also a critical subproblem in many other applications such as the distributed implementation of Lloyd's algorithm for K-means clustering [16] and power iteration for computing eigenvectors [21].

However, the communication complexity of this fundamental problem has not been fully understood, especially when the input is sparse or skew. In this paper, we provide a tight connection between communication complexity and input sparsity. Specifically, we propose a new sparsity-aware lossy compression scheme, which reduces the communication cost both theoretically and empirically. We also prove that the communication cost of our method is information-theoretic optimal up to a constant factor in all sparsity regime, under Hoyer's measure of sparseness [9].

## 1.1 Problem definition and notation

The problem setting in this paper is the same as in [20]. Each client $i$ holds a private vector $X_i \in \mathbb{R}^d$ and transmits messages only to the central sever according to some protocol; at the end, the sever outputs an estimate for the mean $X = \frac{1}{n}\sum_{i=1}^n X_i$ based on all the messages it has received. The communication cost of a protocol is measured by the total number of bits exchanged between clients and the sever. Let $\hat{X}$ denote the estimated mean and we wish to minimize the mean square error (MSE) of the estimate, i.e., $\mathcal{E} = \mathsf{E}\|\hat{X} - X\|_2^2$, under a certain communication budget. Note that the problem considered here is non-stochastic, i.e., the input vectors are arbitrary or even adversarial. This is different from distributed statistical estimation [27, 8, 4], where the inputs are i.i.d samples from some distribution and the goal is to estimate the parameters of the underlying distribution. In particular, the expectation in the above definition of MSE is only over the randomness of the protocol.

We define $F_1 := \sum_{i=1}^n \|X_i\|_1$, i.e., the sum of $\ell_1$-norms of input vectors; let $F_2 := \sum_{i=1}^n \|X_i\|_2^2$ be the sum of squared $\ell_2$-norms of the input vectors and $F_0$ be the total number of non-zero entries in all the input vectors. We will always use $d$ to denote the dimensionality of input vectors and $n$ for the number of clients.

## 1.2 Previous results

Naively sending all vectors to the sever needs $ndr$ bits of communication, where $r$ is the number of bits to represent a floating point number. In [20], several methods to save communication are proposed. The best of them uses $O(nd)$ bits of communication while achieving an MSE of $F_2/n^2$. Their algorithm first applies stochastic quantization and then encodes the quantized vectors by entropy encoding schemes such as arithmetic coding. Moreover, it is also proved that, in the worst case, this cost is optimal for *one-round* protocols. Similar bounds are also obtained in [2, 11]. One major limitation of the methods in [20] is that they cannot exploit the sparseness in the inputs due to the nature of their quantization and encoding methods. In many distributed learning scenarios, the input vectors can be very sparse or skew, i.e., a large fraction of the entries can be zero or close to zero. The sparsity can be caused by either data unbalance (large entries occur in a few clients) or feature unbalance (large entries occur in a few dimensions). QSGD of [2] works well in practice for sparse data, but doesn't have an upper bound on the cost that is parameterized by input sparsity: to achieve an MSE of $F_2/n^2$, the cost is still $O(nd)$ bits (Theorem 3.2, Corollary 3.3 in [2]).

Intuitively, one could drop entries with small absolute values without affecting the MSE too much. Gradient sparsification utilizes this idea, which has been successfully applied in distributed gradient compression [19, 1, 15, 22, 23]. However, such methods either do not have optimal sparsity-sensitive theoretical guarantees or work only under strong sparsity assumptions.

There are various sparsity notions, but it is currently not clear which notion best characterizes the inherent complexity of DME. To get meaningful sparsity-sensitive bounds, it is essential to identify an appropriate sparsity notion for DME. In this paper, we propose to use a modified notion of Hoyer's to measure the sparseness of vectors [9]. For a $d$-dimensional vector $X$, its sparseness is defined as $\frac{\|X\|_1^2}{d\|X\|_2^2}$.[1] Since our inputs can be viewed as an $nd$-dimensional vector, the global sparseness is defined as $s := F_1^2/ndF_2$. Note that $\frac{1}{nd} \le s \le 1$; $s = 1$ (densest) iff all entries are non-zero and have equal absolute values, and $s = \frac{1}{nd}$ (sparsest) iff the input contains a single non-zero entry. Wangni et al. [23] obtain a sparsity-aware bound based on a different sparsity notion, but our result implies theirs and can be much better for some inputs (see the supplementary for details).

## 1.3 Our contributions

First, we propose a sparsity-sensitive compression method that provably exploits the sparseness of the input. Specifically, to achieve an MSE of $\mathcal{E} \le \frac{F_2}{n^2}$, our protocol only needs to transmit $\mathcal{C} \approx nds \log\left(\frac{1}{s} + 1\right)$ bits (ignoring some lower order terms), where $s$ is the sparseness of the input defined earlier. Since $s \log\left(\frac{1}{s} + 1\right) \le 1$ when $s \le 1$, this is always no worse than $nd$ (the cost of [20]) and can be much smaller on sparse inputs, i.e., when $s \ll 1$.

Secondly, we prove that, for any sparseness $s \le 1$, the communication cost of our protocol is optimal, up to a constant factor. Specifically, for any $s \le 1$, we construct a family of inputs with

sparseness equal to $s$, and prove that any protocol achieving an MSE of $\frac{F_2}{n^2}$ on this family must incur $\Omega(nds \log \frac{1}{s})$ bits of communication in expectation for some inputs in this family. This lower bound holds for multi-round protocols in the broadcasting model (where each message can be seen by all clients). As observed in [20], any lower bound for distributed statistical mean estimation can be translated to a lower bound for the DME problem. However, current lower bounds in this area do not suffice to obtain tight sparsity-sensitive bounds for DME.

Finally, we complement our theoretical findings with experimental studies. Empirical results show that, under the same communication bandwidth, our proposed method has a much lower MSE, especially on sparse inputs, which verifies our theoretical analyses. Moreover, as a subroutine, our protocol outperforms previous approaches consistently in various distributed learning tasks, e.g., Lloyd's algorithm for K-means clustering and power iteration.

## 2 Sparsity-Sensitive DME Protocol

**Overview of our techniques.** Algorithms in [20] apply $k$-level stochastic quantization and then encode the quantized vectors using variable length coding. Specifically, for each $X_i$, the client divides the range $[X_i^{\min}, X_i^{\max}]$ into $k-1$ intervals of equal length, and then identifies the interval containing each $X_{ij}$ and rounds it either to the left point or the right point of the corresponding interval with probability depending on its distance to the end points. After quantization, the vector can be viewed as a string of length $d$ over an alphabet of size $k$, which is then compressed using arithmetic coding. QSGD is similar, but encodes signs separately and uses the Elias coding method.

Since the sparseness depends on the $\ell_1$ norm $F_1$, our quantization step size depends on $F_1$ as in Wang et al. [22]. In addition to $F_1$ quantization, our protocol has the following algorithmic ingredients. 1) All clients in our protocol use the same interval size in stochastic quantization. This means that the number of levels may vary for different clients, as opposed to all previous methods, where all clients use a fixed number of levels. This is another major difference in our quantization step, which is necessary to get communication bounds in terms of global sparsity. 2) As in QSGD, we encode the sign of each entry separately and only quantize the absolute values, which can be conveniently implemented by a scaling and rounding procedure. 3) Instead of encoding the quantized vectors directly with entropy coding methods, we first convert each integer vector into a bit string using a one-to-one map: for any integer vector $\boldsymbol{v} = (v_1, v_2, \cdots, v_d)$, the length of its corresponding bit string is $d + \|\boldsymbol{v}\|_1 - 1$, among which the number of 1's is exactly $d - 1$. 4) We then apply efficient coding methods, e.g., arithmetic coding, to encode the entire bit string using roughly $\log \binom{d + \|\boldsymbol{v}\|_1}{d} \approx \|\boldsymbol{v}\|_1 \log \frac{d + \|\boldsymbol{v}\|_1}{\|\boldsymbol{v}\|_1}$ bits.

**Scaling and Rounding.** We first introduce the scaling and rounding procedure (Algorithm 1), which is essentially equivalent to stochastic quantization (for the absolute values only). The next Lemma summarizes the key properties of SaR, the proof of which is in the supplementary.

---

**Algorithm 1** Scaling and Rounding (SaR)

---

**input** $\boldsymbol{v} \in \mathbb{R}^d$ and a scaling factor $F$
1: $\boldsymbol{u} = \frac{1}{F} \cdot \boldsymbol{v}$
2: Randomized rounding: for $j = 1, \cdots, d$

$$\hat{u}_j = \begin{cases} \lfloor u_j \rfloor + 1, & \text{with probability } u_j - \lfloor u_j \rfloor \\ \lfloor u_j \rfloor, & \text{otherwise.} \end{cases}$$

3: **return** $\hat{\boldsymbol{u}}$

---

**Lemma 2.1.** *Let $\hat{\boldsymbol{v}} = F\hat{\boldsymbol{u}}$, then $\mathsf{E}[\hat{\boldsymbol{v}}] = \boldsymbol{v}$ and $\mathsf{E}[\|\hat{\boldsymbol{v}} - \boldsymbol{v}\|_2^2] \leq F\|\boldsymbol{v}\|_1$. Moreover, $\mathsf{E}[|\hat{v}_i|] = |v_i|$.*

In our protocol, we apply Algorithm 1 on each $X_i$ with $F = F_1/C$, where $C$ is a tunable parameter. Let $\hat{\boldsymbol{u}_i}$ be the output for $X_i$ and $\hat{X}_i = F\hat{\boldsymbol{u}_i}$. At the end, the server uses $\hat{X} = \sum_{i=1}^n \hat{X}_i/n$ as the estimate for the mean, then by Lemma 2.1, the MSE is

$$\mathcal{E} = \mathsf{E} \|\hat{X} - X\|_2^2 = \frac{1}{n^2} \sum_{i=1}^n \mathsf{E}[\|\hat{X}_i - X_i\|_2^2] \leq \frac{F}{n^2} \sum_{i=1}^n \|X_i\|_1 = \frac{F_1^2}{Cn^2}. \tag{1}$$

**Constant-weight binary sequence.** The Hamming weight of a length-$d$ binary sequence $\boldsymbol{v}$ is denoted by $w(\boldsymbol{v}) = |\{v_i : v_i = 1\}|$. Constant-weight binary codes $\mathcal{C}(d, w)$ is the set of all length-$d$ sequences with weight $w$. Since $|\mathcal{C}| = \binom{d}{w}$, the number of bits to represent a sequence in $\mathcal{C}$ is at least $\lceil \log \binom{d}{w} \rceil$. There exists efficient encoding methods, such as arithmetic coding or its variants [17], to encode sequences in $\mathcal{C}$ using binary strings of length very close to $\lceil \log \binom{d}{w} \rceil$, which has encoding and decoding time $O(d)$.

**Constant-weight non-negative integer vector coding.** Denote the weight of a length-$d$ non-negative integer vector $\boldsymbol{v}$ by $w(\boldsymbol{v}) = \sum_{i=1}^{d} v_i$. Constant-weight integer codes $\mathcal{I}(d, w)$ is the set of length-$d$ non-negative integer vectors with weight $w$. In our protocol, we map each $\boldsymbol{v} \in \mathcal{I}(d, w)$ to a binary sequence $f(\boldsymbol{v}) \in \mathcal{C}(d + w - 1, d - 1)$ as follows. For $i = 1, 2, \cdots, d - 1$ we write $v_i$ '0's and one '1', and in the end we write $v_d$ '0's. It is an $(d + w - 1, d - 1)$ constant-weight binary code. One can also verify that $f$ is a one-to-one and onto map from $\mathcal{I}(d, w)$ to $\mathcal{C}(d + w - 1, d - 1)$. By applying encoding methods for $\mathcal{C}$ mentioned above, we have the following lemma.

**Lemma 2.2.** *Sequences in $\mathcal{I}(d, w)$ can be encoded losslessly by $\lceil \log \binom{d+w-1}{d-1} \rceil$-bit binary strings, with encoding and decoding time $O(d + w)$.*

## 2.1 The Protocol

We are now ready to describe our sparsity-sensitive DME protocol.

**1. (Initialization)** Clients and the server determine the scaling factor $F$ to be used in Algorithm 1 and we will use $F = F_1/C$ for some $C \le nd$. To compute $F_1$, each client $i$ sends $\|X_i\|_1$ to the server using $r$ bits, where $r$ is the number of bits to represent floating points. Then the server computes $F_1 = \sum_i \|X_i\|_1$ and broadcasts it to all the clients. This step use $2r$ bits of communication per client.

**2. (Quantization)** Client $i$ runs $\mathsf{SaR}(X_i, F_1/C)$ (Algorithm 1) and get an integer vector $\hat{\boldsymbol{u}}_i$. The absolute value and sign of each entry in $\hat{\boldsymbol{u}}_i$ will be encoded separately. Let $\boldsymbol{v}_i = (|\hat{\boldsymbol{u}}_{i1}|, \cdots, |\hat{\boldsymbol{u}}_{id}|)$.

**3. (Encoding)** Note that $\boldsymbol{v}_i \in \mathcal{I}(d, w_i)$, where $w_i = w(\boldsymbol{v}_i)$. Client $i$ encodes $\boldsymbol{v}_i$ using a $\lceil \log \binom{d+w_i-1}{d-1} \rceil$-bit string (Lemma 2.2) and sends it to the sever. The client also sends the value $\Delta w_i = w_i - \lfloor C\|X_i\|_1/F_1 \rfloor$ with $\log(2d+1)$ bits [2], as $w_i$ is needed for decoding $\boldsymbol{v}_i$. [3]

**4. (Sending the signs)** Let $\boldsymbol{s}_i$ be a binary sequence indicating the signs of non-zero entries in $\hat{\boldsymbol{u}}_i$. Client $i$ simply sends this sequence with $d_i$ bits of communication, where $d_i$ is the number of non-zero values in $\boldsymbol{v}_i$. Moreover, we can apply constant-weight coding to compress this sequence.

**5. (Decoding)** The server decodes $\boldsymbol{v}_i$, which contains the absolute values of $\hat{\boldsymbol{u}}_i$. Given the signs of its the non-zero entries $\boldsymbol{s}_i$, the server is now able to recover $\hat{\boldsymbol{u}}_i$ losslessly. It finally computes the estimated mean $\hat{X} = \frac{1}{n} \sum_i \hat{X}_i = \frac{F_1}{Cn} \sum_i \hat{\boldsymbol{u}}_i$.

The correctness of the protocol readily follows from Lemma 2.1 and (1): $\mathsf{E}[\hat{X}] = X$ and $\mathsf{E}[\|\hat{X} - X\|_2^2] \le \frac{F_1^2}{Cn^2}$. Below we analyze its communication cost. By part 2 of Lemma 2.1, we have $\mathsf{E}[w_i] = \sum_j \mathsf{E}[|\hat{u}_{ij}|] = \sum_j \frac{C|X_{ij}|}{F_1} = \frac{C\|X_i\|_1}{F_1}$. Therefore, $\mathsf{E}[\sum_{i=1}^{n} w_i] = \sum_{i=1}^{n} \frac{C\|X_i\|_1}{F_1} = C$.

Because of the observation $d_i \le w_i$, we have the expected total communication cost is at most

$$\mathsf{E}\left[ \sum_{i=1}^{n} \left( 2r + \log(2d+1) + \log \binom{d+w_i-1}{d-1} + d_i \right) \right]$$

$$\le \mathsf{E}\left[ \sum_{i=1}^{n} w_i \log\left( \frac{d}{w_i} + 1 \right) \right] + O(C + nr + n \log d).$$

From the concavity of the function $x \log(\frac{1}{x} + 1)$ on $\mathbb{R}_{>0}$ and Jensen's inequality, we have

$$\mathsf{E}\left[\sum_{i=1}^{n} w_i \log\left(\frac{d}{w_i} + 1\right)\right] \leqslant \mathsf{E}\left[(\sum_{i=1}^{n} w_i) \log\left(\frac{nd}{\sum_{i=1}^{n} w_i} + 1\right)\right] \leqslant C \log\left(\frac{nd}{C} + 1\right).$$

Therefore, we get the following theorem, and by setting $C = F_1^2/F_2$, the next corollary follows.

**Theorem 2.3.** *For any $C \leq nd$, there exists a DME protocol that achieves an MSE of $\frac{F_1^2}{Cn^2}$ with $C \log\left(\frac{nd}{C} + 1\right) + O(C + nr + n \log d)$ bits of communication, where $r$ is the number of bits to represent a floating point.*

**Corollary 2.4.** *There exists a DME protocol that achieves an MSE of $\frac{F_2}{n^2}$ using $nds \cdot \log\left(\frac{1}{s} + 1\right) + O(nds + nr + n \log d)$ bits, where $s = F_1^2/ndF_2$ is the Hoyer's measure of sparseness of the inputs.*

**Remark.** The authors of [20] discuss how to use client or coordinate sampling to obtain a trade-off between MSE and communication. Their analysis shows that, to achieve an MSE of $F_2/pn^2$, the communication cost is $O(pnd)$ bits (ignoring low order terms), where $0 \leq p \leq 1$ is the sampling probability. Applying our sparsity-sensitive protocol on the sampled clients or coordinates, we can achieve the same MSE with a communication of $O(pnds \log(\frac{1}{s} + 1))$ bits, which will never be worse and can be much better on inputs with small spareness $s$.

We also would like to point out that, our algorithm can also be run without the synchronization round. For this setting, we can derive a communication bound for each client by simply setting $n = 1$ in Corollary 2.4, although $s$ in the bound will become the local sparsity of the client when doing so. Local sparsity bound is worse than global sparsity when there is data unbalance, but the bound is still better than prior work as long as there is dimension unbalance across different clients. This is also verified in our experimental results below.

Since the sparseness depends on the $\ell_1$ norm $F_1$, the key to getting a sparsity-sensitive bound is to understand the connection between $F_1$ and the MSE-communication trade-off. So our quantization step size depends on $F_1$, which is one of the main differences in the quantization step compared with [20, 2, 23]. Wang et al. [22] also use $F_1$ quantization, but only consider 1-level quantization and doesn't specify an appropriate encoding method to achieve an optimal sparsity-sensitive bound. Our protocol uses $C \log\left(\frac{nd}{C} + 1\right)$ bits of communication to achieve an MSE of $F_1^2/n^2C$, where $C \leq nd$ is a tunable parameter; and if we set $C = F_1^2/F_2$, the MSE and communication cost are $F_2/n^2$ and $nds \log(1/s + 1)$ respectively, as claimed earlier. Having $C$ as a tunable parameter gives us a better control on the cost of the protocol; our result in fact implies the MSE-communication trade-off of [22] (and could be much better) but not vice versa. Wang et al. [22] prove that their algorithm can compress a vector $X \in \mathbb{R}^d$ using $kr$ bits (where $r$ is the number of bits to represents floating points and $k$ is a tuning parameter) with MSE $F_1^2/k$; ours algorithm (the special case when $n = 1$) can compress $X$ using $C \log(\frac{d}{C} + 1)$ bits with MSE at least $F_1^2/C$. By setting $C = k$, we achieve the same MSE while the cost is $k \log(d/k)$ bits (and it is trivial to make it be $k \min(\log(d/k), r)$). When $k = \Theta(d)$, the cost is $O(k)$ bits versus $O(kr)$.

## 3 Lower Bound

In this section, we show the optimality of Theorem 2.3 by proving the following lower bound.

**Theorem 3.1.** *For any $n \leq C \leq \frac{nd}{2}$, there exists a family of inputs, all of which have $F_1 = F_2 = C$, such that any randomized protocol solving the DME problem on this family in the broadcast model with an MSE of $\frac{F_1^2}{4n^2C}$ must communicate at least $\frac{C}{2} \log \frac{nd}{2C}$ bits.*

This theorem immediately leads to the following corollary, which means that our sparsity-sensitive protocol is optimal (up to a constant factor) for all sparseness $\frac{1}{d} \leq s \leq \frac{1}{2}$.

**Corollary 3.2.** *For any sparseness $\frac{1}{d} \leq s \leq \frac{1}{2}$, there exists a family of inputs, all of which have sparseness $s$, such that any randomized protocol solving the DME problem on this family in the broadcast model with an MSE of $\frac{F_2}{4n^2}$ must communicate at least of $\frac{nds}{2} \log(\frac{1}{2s})$ bits.*

*Proof.* Note that on the family of inputs used in the proof of Theorem 3.1 (presented shortly), we have $s = \frac{F_1^2}{ndF_2} = \frac{C}{nd}$ and $\mathcal{E} = \frac{F_1^2}{4n^2C} = \frac{F_2}{n^2}$. Since this family exists for any $n \leq C \leq \frac{nd}{2}$, we obtain a

family with sparseness $s$ for any $\frac{1}{d} \leq s \leq \frac{1}{2}$. Then, the MSE and communication cost in Theorem 3.1 can be rewritten as claimed in the corollary. □

The rest of this section is devoted to the proof of Theorem 3.1. To prove lower bounds for randomized protocols, the standard tool is Yao's *Minimax Principle* [25]. We will define an input distribution $\mathcal{D}$ for DME. Suppose there is a randomized algorithm $\mathcal{A}_R$ with worst case (for any possible input) MSE $M$ and expected cost $T$, where $R$ is the randomness used in the algorithm. Now, if we sample input $X \sim \mathcal{D}$, then $\mathsf{E}_R \mathsf{E}_{X \sim \mathcal{D}}[\text{MSE of } \mathcal{A}_R(X)] \leq M$ and $\mathsf{E}_R \mathsf{E}_{X \sim \mathcal{D}}[\text{Cost of } \mathcal{A}_R(X)] \leq T$. By Markov's inequality, $\mathsf{Pr}_R [\mathsf{E}_{X \sim \mathcal{D}}[\text{ MSE of } A_R(X)] \leq 4M] \geq 0.75$ and $\mathsf{Pr}_R [\mathsf{E}_{X \sim \mathcal{D}}[\text{Cost of } A_R(X)] \leq 2T] \geq 0.5$. Then, with positive probability, the two events happen simultaneously. In other words, there exists some fixed randomness $r$ such that $\mathsf{E}_{X \sim \mathcal{D}}[\text{ MSE of } A_r(X)] \leq 4M$ and $\mathsf{E}_{X \sim \mathcal{D}}[\text{Cost of } A_r(X)] \leq 2T$ where $A_r$ is now simply a deterministic algorithm. That means if there is a randomized algorithm with worst case MSE $M$ and expected cost $T$, then there exist a deterministic algorithm with MSE $4M$ and expected cost $2T$ w.r.t. any input distribution $\mathcal{D}$.

**Minimax Principle.** From the above argument, it is sufficient to prove that, for some input distribution $\mathcal{D}$, any *deterministic* protocol with MSE at most $\Theta(F_1^2/n^2 C)$ must incur an expected communication cost of $\Omega(C \log \frac{nd}{C})$ bits.

**Input distribution.** For any fixed $n \leq C \leq nd/2$, we define the hard distribution $\mathcal{D}$ for our problem as follows. Each $X_i$ is divided into $t = C/n$ blocks, each of size $b = nd/C$. In this section, we use $x_{ij} \in \mathbb{R}^b$ to denote the $j$th block in $X_i$. In $\mathcal{D}$, each block $x_{ij}$ is uniformly sampled from $b$-dimensional standard basis vectors, i.e. $\mathsf{Pr}[x_{ij} = e_k] = 1/b$ for each $1 \leq k \leq b$, and the distribution of $x_{ij}$ are *independent* across all $i$ and $j$. Note that any input sampled from $\mathcal{D}$ has $\ell_1$ norm exactly $C$.

Let $\Pi$ be any *deterministic* protocol with MSE bounded by $F_1^2/4n^2 C = C/4n^2$ w.r.t. the input distribution $\mathcal{D}$. We next prove a lower bound of the expected communication cost of $\Pi$ w.r.t. $\mathcal{D}$. Let $X_1, X_2, \cdots, X_n$ be a random input sampled from $\mathcal{D}$ and $\Pi(X_1, \cdots, X_n)$ be the transcript of the protocol given the input, i.e., the concatenation of all messages, which is a random variable. When there is no confusion, we will omit the input and use $\Pi$ to denote the random transcript; and $\pi \sim \Pi$ means $\pi$ is chosen according to the distribution of $\Pi(X_1, \cdots, X_n)$.

Since the protocol is deterministic, any particular transcript $\pi$ corresponds to a deterministic set of inputs $R_\pi$, i.e., all inputs in $R_\pi$ generate the same transcript $\pi$ under the protocol $\Pi$. Hence, all inputs in $R_\pi$ share the same output, denoted as $Y^\pi$. Note each input belongs to a unique $R_\pi$, and thus all $R_\pi$ corresponds to a partition of all possible inputs. It is well-known $R_\pi$ is a combinatorial rectangle, i.e., $R_\pi = B_1 \times \cdots \times B_n$, where each $B_i \subseteq \{0,1\}^d$ is some subset of all possible inputs of the $i$th client.

**Definition 1.** *Define $\mathcal{D}_\pi$ as the conditional distribution of $X_1, X_2, \cdots, X_n$ (sampled from $\mathcal{D}$) conditioned on the event $[X_1, X_2, \cdots, X_n] \in R_\pi$.*

Let $X = \frac{1}{n} \sum_{i=1}^n X_i$. By the property of conditional expectation, we have the following Lemma.

**Lemma 3.3.** *We assume $\Pi$ has an MSE of $\frac{C}{4n^2}$, then $\mathsf{E}_{\pi \sim \Pi} \mathsf{E}_{[X_1, \cdots, X_n] \sim \mathcal{D}_\pi} \left[ \|X - Y^\pi\|^2 \right] \leq \frac{C}{4n^2}$.*

**Definition 2.** *Suppose $[X_1, \cdots, X_n] \sim \mathcal{D}_\pi$, then for every $i$ and $j$, the distribution of $x_{ij}$ is still a distribution over $b$-dimensional basis vectors. For each $i, j$, we define $p_{ijk}^\pi = \mathsf{Pr}_{\mathcal{D}_\pi}[x_{ij} = e_k]$ for $k \in [b]$, where $\sum_{k=1}^b p_{ijk}^\pi = 1$.*

The next lemma is crucial to our argument, the proof of which can be found in the supplementary.

**Lemma 3.4.** *For any $\pi$ and let $Y^\pi$ be its output, we have*

$$\sum_{j=1}^t \sum_{k=1}^b \sum_{i=1}^n [p_{ijk}^\pi (1 - p_{ijk}^\pi)] \leq n^2 \cdot \mathsf{E}_{[X_1, \cdots, X_n] \sim \mathcal{D}_\pi} \left[ \|X - Y^\pi\|^2 \right].$$

Here we introduce some basic notations from information theory [6]. For any random variable $X$, $H(X)$ is the standard Shannon Entropy of $X$. For any random variables $X, Y, Z$, we use

$H(X|Y) = \mathsf{E}_Y[H(X|Y = y)]$ to denote the conditional entropy of $X$ given $Y$, and $I(X;Y|Z) = H(X|Z) - H(X|Y, Z)$ to denote the conditional mutual information between $X$ and $Y$ given $Z$. We know the average encoding length of a random transcript $\Pi$, i.e., the expected communication cost, is lower bounded by its entropy $H(\Pi)$. By the non-negativity of (conditional) entropy, we have

$$H(\Pi) = I(X_1, \cdots, X_n; \Pi) + H(\Pi|X_1, \cdots, X_n) \geq I(X_1, \cdots, X_n; \Pi). \tag{2}$$

Next we prove a lower bound on $I(X_1, \cdots, X_n; \Pi)$. We will need the following property.

**Lemma 3.5.** *Let $X, Y, Z$ be three random variables such that $X$ and $Y$ are independent, then $I(X, Y; Z) \geq I(X; Z) + I(Y; Z)$.*

**Lemma 3.6.** $I(X_1, \cdots, X_n; \Pi) \geq \frac{C}{2} \log \frac{nd}{2C}$.

*Proof.* Since the input distribution is independent across different blocks and clients, by Lemma 3.5, we have $I(X_1, \cdots, X_n; \Pi) \geq \sum_{j=1}^{t} \sum_{i=1}^{n} I(X_{ij}; \Pi)$. Thus,

$$I(X_1, \cdots, X_n; \Pi) \geq \sum_{j=1}^{t} \sum_{i=1}^{n} H(X_{ij}) - \sum_{j=1}^{t} \sum_{i=1}^{n} H(X_{ij} \mid \Pi)$$

$$= C \log \frac{nd}{C} - \mathsf{E}_{\pi \sim \Pi} \left[ \sum_{j=1}^{t} \sum_{i=1}^{n} H(X_{ij} \mid \Pi = \pi) \right], \tag{3}$$

where we use $H(X_{ij}) = \log b = \log \frac{nd}{C}$. Let $q_{ij}^{\pi} = \min(p_{ij}^{\pi}, 1 - p_{ij}^{\pi})$ (see Definition 2), then $q_{ij}^{\pi} \leq 0.5$. It can be verified by elementary calculus that $q_{ij}^{\pi} \log \frac{1}{q_{ij}^{\pi}} \geq (1 - q_{ij}^{\pi}) \log \frac{1}{1 - q_{ij}^{\pi}}$, which implies that $q_{ij}^{\pi} \log \frac{1}{q_{ij}^{\pi}} \geq p_{ij}^{\pi} \log \frac{1}{p_{ij}^{\pi}}$. So,

$$\mathsf{E}_{\pi \sim \Pi} \left[ \sum_{j=1}^{t} \sum_{i=1}^{n} H(X_{ij} \mid \Pi = \pi) \right] = \mathsf{E}_{\pi \sim \Pi} \left[ \sum_{j=1}^{t} \sum_{i=1}^{n} \sum_{k=1}^{b} \left( p_{ijk}^{\pi} \log \frac{1}{p_{ijk}^{\pi}} \right) \right]$$

$$\leq \mathsf{E}_{\pi \sim \Pi} \left[ \sum_{j=1}^{t} \sum_{i=1}^{n} \sum_{k=1}^{b} \left( q_{ijk}^{\pi} \log \frac{1}{q_{ijk}^{\pi}} \right) \right]$$

$$\leq \mathsf{E}_{\pi \sim \Pi} \left[ (\sum_{j=1}^{t} \sum_{i=1}^{n} \sum_{k=1}^{b} q_{ijk}^{\pi}) \log \frac{nd}{\sum_{j=1}^{t} \sum_{i=1}^{n} \sum_{k=1}^{b} q_{ijk}^{\pi}} \right]$$

$$\leq \mathsf{E}_{\pi \sim \Pi} [\sum_{j=1}^{t} \sum_{i=1}^{n} \sum_{k=1}^{b} q_{ijk}^{\pi}] \log \frac{nd}{\mathsf{E}_{\pi \sim \Pi} [\sum_{j=1}^{t} \sum_{i=1}^{n} \sum_{k=1}^{b} q_{ijk}^{\pi}]}$$

where the last two inequalities is from Jensen's inequality (since $x \log(1/x)$ is concave on $\mathbb{R}_{>0}$).

Since each $q_{ijk}^{\pi} \leq 0.5$ and by Lemma 3.3 and 3.4, we have

$$\mathsf{E}_{\pi \sim \Pi} \left[ \sum_{j=1}^{t} \sum_{i=1}^{n} \sum_{k=1}^{b} q_{ijk}^{\pi} \right] \leq 2\mathsf{E}_{\pi \sim \Pi} \left[ \sum_{j=1}^{t} \sum_{i=1}^{n} \sum_{k=1}^{b} [q_{ijk}^{\pi}(1 - q_{ijk}^{\pi})] \right]$$

$$= 2\mathsf{E}_{\pi \sim \Pi} \left[ \sum_{j=1}^{t} \sum_{i=1}^{n} \sum_{k=1}^{b} [p_{ijk}^{\pi}(1 - p_{ijk}^{\pi})] \right] \leq \frac{C}{2}.$$

Consider the function $g(x) = x \log \frac{nd}{x}$, which is concave on $\mathbb{R}_{>0}$ and its derivative is $g'(x) = \log \frac{nd}{x} - \frac{1}{\ln 2}$. Thus $g(x)$ attains its maximum at $x = \frac{nd}{2^{1/\ln 2}}$. Moreover, $g(x)$ is monotonically increasing for $0 < x \leq \frac{nd}{2^{1/\ln 2}}$. Since we assume $\frac{C}{2} \leq \frac{nd}{4} < \frac{nd}{2^{1/\ln 2}}$, we have $\mathsf{E}_{\pi \sim \Pi} \left[ \sum_{j=1}^{t} \sum_{i=1}^{n} H(X_{ij} \mid \Pi = \pi) \right] \leq \frac{C}{2} \log \frac{2nd}{C}$. By (3), we prove

$$I(X_1, \cdots, X_n; \Pi) \geq C \log \frac{nd}{C} - \frac{C}{2} \log \frac{2nd}{C} = \frac{C}{2} \log \frac{nd}{C} - \frac{C}{2} = \frac{C}{2} \log \frac{nd}{2C}.$$

This finishes the proof of the Lemma. $\qquad\square$

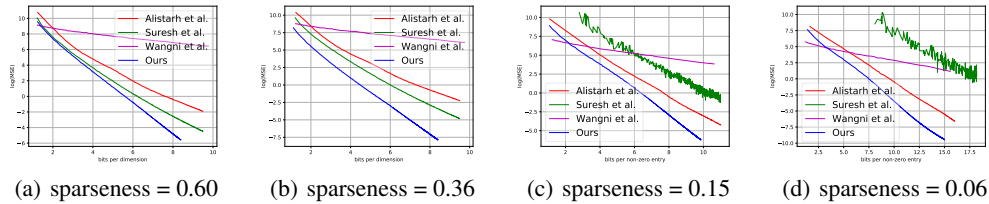

| (a) sparseness = 0.60 | (b) sparseness = 0.36 | (c) sparseness = 0.15 | (d) sparseness = 0.06 |

Figure 1: Communication-MSE trade-off on the synthetic dataset generated from t-distribution. The x-axis is the average number of bits sent for each dimension, and the y-axis is log(MSE).

By (2), we prove that the expected communication cost of $\Pi$ is at least $\frac{C}{2} \log \frac{nd}{2C}$ bits w.r.t. $\mathcal{D}$. Then Theorem 3.1 follows from the minimax principle.

## 4 Experiments

We have conducted experiments comparing our DME protocol with the variable length coding method (the best in [20]) and the methods in [2, 23] on their MSE-communication trade-off, as well as the performance in distributed learning tasks that use DME as a subroutine, including K-means clustering and power iteration. The algorithm in [22] doesn't specify an appropriate encoding method and directly sends floating points, and thus the cost is worse than that of [2, 23].

### 4.1 DME

In the first set of experiments, we compare our new protocol with that of [2, 23, 20] on the DME problem directly, in terms of the MSE-communication trade-off. To see how the performances of the protocols are affected by the sparseness of the input, we generated synthetic datasets with varying spareness. Specifically, we generated 16 vectors, each held by a different client. Each vector has 10000 dimensions, whose values are generated independently from student's t-distribution. This data set has an empirical sparseness of 0.60, and the results are shown in Figure 1(a).

We used two ways to create sparser data. First, we scaled up the data on each nodes by a different factor, which is also generated from t-distribution. This resulted in a data set with sparseness 0.36, and the experimental results are shown in Figure 1(b), which confirms the effectiveness of using a global quantization step size when data is unbalance across clients. Second, we randomly chose 30% and 10% of the dimensions and set the rest to 0. This resulted in two data sets with sparseness 0.15 and 0.06, respectively, and the experimental results are shown in Figure 1(c) and Figure 1(d). These results render that the sparser and/or less balance (across clients) the data is, the higher performance gain our new protocol has. The same phenomenon is also observed in the next two tasks.

In Figure 1 (a)(c)(d), the data sets used do not have data unbalance across different clients (meaning the coordination round is effectively useless), and the results are still better than previous methods.

### 4.2 Distributed K-Means

We then test the performances for distributed K-means. In each iteration of the distributed K-means algorithm, the server broadcasts the current centroids of the clusters to all clients. Each client updates the centroids based on its local data, and then sends back the updated centroids to the server. The server then computes the average of these centroids for each cluster. This is exactly $K$ instances of the DME problem, except that average should be weighted by the cluster size at each client. Thus, we first scale up the centroids by the cluster size, and then apply the DME protocols.

We used the MNIST [13] data set, uniformly or non-uniformly distributed across 10 clients. The number of clusters and iterations is set to 10 and 30 respectively. The results are shown in Figure 2, where we used different values of $k$ (quantization level) for Suresh et al.'s algorithm, $k = 32$ for less communication and $k = 512$ for less error, and other methods are tuned to achieve the same objective. The results show that with the same final objective, our algorithm has less communication cost.

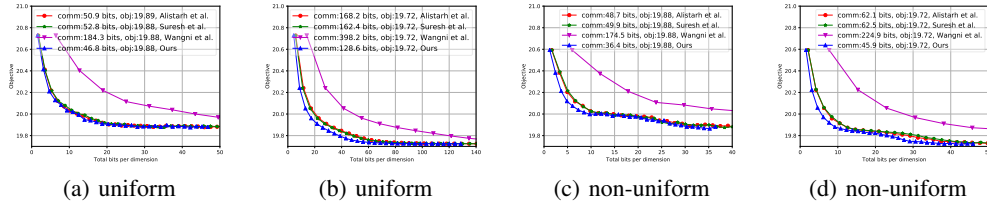

| (a) uniform | (b) uniform | (c) non-uniform | (d) non-uniform |

Figure 2: Distributed K-Means on MNIST dataset distributed among 10 workers. The x-axis is the average number of bits sent for each dimension, accumulated over the iterations, and the y-axis is the objective function value of K-Means. In (a) and (b) data is uniform distributed, while in (c) and (d) data is non-uniform distributed, every worker has 1000, 4000, 7000, 10000 or 13000 images.

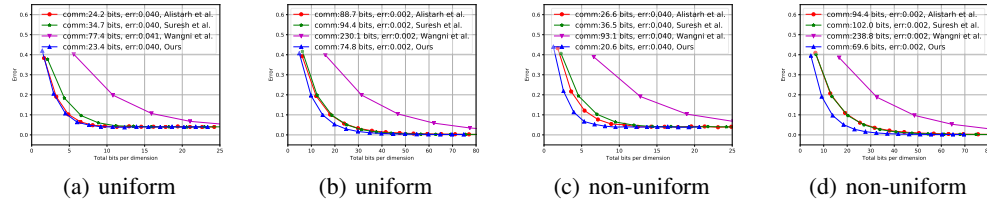

| (a) uniform | (b) uniform | (c) non-uniform | (d) non-uniform |

Figure 3: Distributed power iteration on MNIST dataset distributed among 100 workers. The x-axis is the averaged number of bits sent for each dimension, which scales linearly to the number of iterations, and the y-axis is the $\ell_2$ distance between the current estimate of eigenvector and the ground-truth eigenvector. In (a) and (b) data is uniform distributed, while in (c) and (d) data is non-uniform distributed, every worker has 100, 400, 700, 1000 or 1300 images.

## 4.3 Distributed Power Iteration

The second learning task we tested is the distributed power iteration algorithm. The number of clients is set to 100 and the number of iterations is set to 15. In this algorithm, the server broadcasts the current estimate of the eigenvector to all clients, then each client updates the eigenvector based on one power iteration on its local data, and sends back the compressed eigenvector to the server. The server updates the current estimate of eigenvector with the average of all the received eigenvectors. The results on the MNIST data set are reported in Figure 3, where we used different values of $k$ (quantization level) for Suresh et al.'s algorithm, $k = 32$ for less communication and $k = 512$ for less error, and other methods are tuned to achieve the same error. It also shows that our DME protocol uses less communication to achieve the same error.

### Acknowledgments

Zengfeng Huang is partially supported by National Natural Science Foundation of China (Grant No. 61802069), Shanghai Sailing Program (Grant No. 18YF1401200) and Shanghai Science and Technology Commission (Grant No. 17JC1420200). Ziyue Huang, Yilei Wang, and Ke Yi are supported by HKRGC under grants 16200415, 16202317, and 16201318.

## Footnotes

[1]The original Hoyer's measure is the ratio between the $\ell_1$ and $\ell_2$ norm, normalized to the range $[0, 1]$.

[2]Clearly, $\Delta w_i$ is an integer and $|\Delta w_i| \le d$. One can also use universal code such as Elias gamma code [7] to reduce the bits of transmitting $\Delta w_i$.

[3]One can also use entropy coding to encode $\boldsymbol{v}_i$, but it is unclear whether such methods achieve the same theoretical guarantee as ours.

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
