[Supplementary Material]

# Supplementary

## 1 Comparison of Our Upper Bound with that of Wangni et al. [2]

Wangni et al. [2] also define a sparsity notion (Definition 2). They call a vector a $X$ $(\rho, k)$-approximately sparse, if the $\ell_1$ norm of $X$ excluding the top $k$ entries is at most $\rho$ times the $\ell_1$ norm of the top $k$ entries. They show for such a vector, with communication cost roughly $O(k \log d + k\rho \log d)$ bits (Theorem 4), the MSE (thus the convergence rate) blows up by $(1 + \rho)$. This result is strictly worse than ours. Consider the $d$ dimensional vector $X = (d^{0.9}, 1, 1, \cdots, 1)$. This vector is $(1, 0.5d)$-approximately sparse for large enough $d$, so their cost is $O(d \log d)$. However, according to our notion of sparsity, the Hoyer's sparseness is $d^{-0.8}$, and thus the communication cost is $d^{0.2} \log d^{0.8}$, which is asymptotically much better. On the other hand, for $(\rho, k)$-approximately sparse vector, its Hoyer's sparseness is at most $(1 + \rho)^2 k/d$. For the most interesting case $\rho = O(1)$, our results implies their bound up to a constant, but not vice versa. For large $\rho$, their cost is $O(\rho k \log d)$ and MSE is $\rho F_2/n^2$. For our algorithm, we can use coordinate sampling as in [1] and achieve the same MSE with cost $O(\rho k \log \frac{d}{k})$, which also implies their result. To sum up, the result of [2] is implied by ours up to a constant, but there exist input instances such that our bound is asymptotically much better.

## 2 Missing Proofs

### 2.1 Proof of Lemma 2.1

**Lemma 2.1** (Lemma 2.1 restated). *Let* $\hat{\boldsymbol{v}} = F\hat{\boldsymbol{u}}$, *then* $\mathsf{E}[\hat{\boldsymbol{v}}] = \boldsymbol{v}$ *and* $\mathsf{E}[\|\hat{\boldsymbol{v}} - \boldsymbol{v}\|_2^2] \leq F\|\boldsymbol{v}\|_1$. *Moreover,* $\mathsf{E}[|\hat{v}_i|] = |v_i|$.

*Proof.* One can verify that $\mathsf{E}[\hat{v}_j] = v_j$ (thus $\mathsf{E}[\hat{\boldsymbol{v}}] = \boldsymbol{v}$). Also, $\mathsf{E}[(\hat{u}_j - u_j)^2] = (u_j - \lfloor u_j \rfloor)(\lfloor u_j \rfloor + 1 - u_j)$. For $u_j \geq 0$, this is bounded by $u_j - \lfloor u_j \rfloor \leq u_j$; for $u_j < 0$, this is bounded by $\lfloor u_j \rfloor + 1 - u_j \leq |u_j|$. Thus, for any $u_j$, $\mathsf{E}[(\hat{u}_j - u_j)^2] \leq |u_j|$. We have

$$\mathsf{E}[\|\hat{\boldsymbol{v}} - \boldsymbol{v}\|_2^2] = \sum_{j=1}^{d} \mathsf{E}[(\hat{v}_j - v_j)^2] = F^2 \sum_{j=1}^{d} \mathsf{E}[(\hat{u}_j - u_j)^2] \leq F^2 \sum_{j=1}^{d} |u_j| = F \sum_{j=1}^{d} |v_j| = F\|\boldsymbol{v}\|_1.$$

For the second part, because scaling and rounding doesn't change the sign of each entry,

$$\mathsf{E}[\mathrm{sign}(v_i) \cdot |\hat{v}_i|] = \mathsf{E}[\hat{v}_i] = v_i = \mathrm{sign}(v_i) \cdot |v_i|,$$

which implies $\mathsf{E}[|\hat{v}_i|] = |v_i|$. $\qquad\square$

### 2.2 Proof of Lemma 3.4

**Lemma 2.2** (Lemma 3.4 restated). *For any* $\pi$ *and let* $Y^\pi$ *be its output, we have*

$$\sum_{j=1}^{t} \sum_{k=1}^{b} \sum_{i=1}^{n} [p_{ijk}^\pi (1 - p_{ijk}^\pi)] \leq n^2 \cdot \mathsf{E}_{[X_1, \cdots, X_n] \sim \mathcal{D}_\pi} \left[ \|X - Y^\pi\|^2 \right].$$

24  *Proof.* We need the following lemma, which is a result of the rectangle property and the fact that the
25  inputs sampled from $\mathcal{D}$ are independent across all clients.

26  **Lemma 2.3.** *Let $X_1, X_2, \cdots, X_n$ be a random inputs sampled from $\mathcal{D}_\pi$ for any particular transcript*
27  $\pi$. *Then, $X_1, X_2, \cdots, X_n$ are still independent of each other.*

28  We have

$$
\mathsf{E}_{[X_1,\cdots,X_n]\sim\mathcal{D}_\pi}\left[\|X-Y^\pi\|^2\right] = \mathsf{E}_{[X_1,\cdots,X_n]\sim\mathcal{D}_\pi}\left[\sum_{j=1}^t \|\frac{1}{n}\sum_{i=1}^n X_{ij} - Y_j^\pi\|^2\right]
$$

$$
= \frac{1}{n^2}\mathsf{E}_{[X_1,\cdots,X_n]\sim\mathcal{D}_\pi}\left[\sum_{j=1}^t\sum_{k=1}^b(\sum_{i=1}^n X_{ijk} - nY_{jk}^\pi)^2\right]
$$

$$
= \frac{1}{n^2}\sum_{j=1}^t\sum_{k=1}^b \mathsf{E}_{[X_1,\cdots,X_n]\sim\mathcal{D}_\pi}\left[(\sum_{i=1}^n X_{ijk} - nY_{jk}^\pi)^2\right]
$$

29  By elementary calculus, for any fixed $y$, one can verify $\mathsf{E}[(X-y)^2] \geq \mathsf{E}[(X-\mathsf{E}[X])^2] = \mathsf{Var}[X]$
30  for any random variable $X$. Therefore, we have

$$
\mathsf{E}_{[X_1,\cdots,X_n]\sim\mathcal{D}_\pi}\left[(\sum_{i=1}^n X_{ijk} - nY_{jk}^\pi)^2\right] \geq \mathsf{Var}_{[X_1,\cdots,X_n]\sim\mathcal{D}_\pi}\left[\sum_{i=1}^n X_{ijk}\right]
$$

$$
= \sum_{i=1}^n \mathsf{Var}_{[X_1,\cdots,X_n]\sim\mathcal{D}_\pi}\left[X_{ijk}\right]
$$

$$
= \sum_{i=1}^n p_{ijk}^\pi(1-p_{ijk}^\pi),
$$

where the first equality holds because, when $[X_1,\cdots,X_n]\sim\mathcal{D}_\pi$, $X_{ijk}$'s are independent across $i$
(Lemma 2.3). Combined with the previous equation, we get

$$
\mathsf{E}_{[X_1,\cdots,X_n]\sim\mathcal{D}_\pi}\left[\|X-Y^\pi\|^2\right] \geq \frac{1}{n^2}\sum_{j=1}^t\sum_{k=1}^b\sum_{i=1}^n[p_{ijk}^\pi(1-p_{ijk}^\pi)],
$$

31  which completes the proof. $\qquad\square$