[Reviews · NeurIPS 2019]

Reviewer 1



Originality: The proposed compression scheme involves significant novelty. The paper clearly cites the prior work and contrast its contribution from the prior work. As claimed in the paper, this paper proposes the first compression scheme (along with tight up to constants) that characterizes the effect of underlying sparsity on the communication-cost and MSE trade-off for the DME problem across all sparsity regimes. Quality: As far as the reviewer can tell, the results presented in the paper are correct. The paper is well organized and it definitely constitutes as a complete piece of work. Clarity: The paper is well written and clearly conveys the main ideas. In fact, the paper is able to nicely summarize all the main ingredients of the proofs in the main text. (Please see below for some minor typos.) Significance: The DME problems is of great importance in distributed computing setups as it arises in many tasks as a subroutine, e.g., distributed gradient descent, distributed k-means. The paper makes significant contributions towards this problem when the underlying vectors are sparse (which is also quite prevalent in the practice). Given the number of recent research efforts in the domain, the reviewer believes that this work is certainly going to receive the attention of other researchers. ########## Post-rebuttal ############# I have gone through the response of the authors and other reviewers' comments. I think I am comfortable with my previous score.

Reviewer 2



Positives: - an almost tight answer to a fundamental problem - good practical performance on some tasks - the lower bound argument is quite nice Negatives: - writing and positioning w.r.t. related work can be improved - experiments on training small CNNs or LSTMs with sparsity could be added, but are not absolutely necessary I am quite positive about this paper. My main observation to the authors is to stress less about the relation with previous work, and stress more about explaining your algorithm in detail. For instance, I think the discussion in lines 101-115 which basically re-explains how your algorithm is different to a finer level of detail could be left for *after* you've actually described your algorithm and its guarantees. You could have a whole sub-section for that, where you can carefully cover the similarities and differences. Currently, the draft is very defensive on this point; this makes sense, but notice that you are the first to be proving tight bounds, so it's kind of OK if your algorithm is not super-novel (given the amount of activity in this area one wouldn't really expect that anyway). The main shortcoming of your algorithm though is from the practical side: you need to synch the normalization factor, which means that your algorithm is two-rounds. This could potentially negate the benefits of reduced communication. It would be nice if you could comment on this. The paper has many typos (some collected below), which should be fixed. - maybe add "asymptotically" optimal to the title. sometimes in communication complexity people actually care about constants (see e.g. list decoding) 105 -> doen't 110 -> trade-off of 101 -> 115: maybe move this discussion for *after* you've presented your algorithm? 128 -> summarizes 157-> sends it 181-182: repeated discussion

Reviewer 3



*Originality and Significance* I think the contribution of the current paper is quite incremental over [2,20] and other works on similar topic, some of which the authors have not cited (such as the ATOMO algorithm which provides a method for handling sparse vectors). But more than this, I feel the authors have failed to appreciate a few points: 1. The authors have allowed small interaction between the parties and even downlink communication from the centre to the parties. This communication is critical for normalising the coordinates and using a quantiser with a fixed dynamic range. However, one of the main points in [20] was to rotate randomly so that this range can be determined by \ell_2 norms, which was assumed to be bounded and need not be described. Authors have a rather simple scheme, but it still requires coordination between the parties by the centre. I feel that random rotations such as randomised Hadamard transform should have been used to avoid this. 2. The lower bound seems to only consider SMPs and doesn't allow for the downlink communication by the centre. In this sense, I feel that the optimality of the proposed scheme has not been established in a strict sense. The authors continue with the running assumption of [2,20] that the norm can be described using a fixed number of bits that can be ignored. I find this assumption a bit absurd for theoretical contributions, but am now accustomed to seeing it in papers on this literature. However, the authors should clearly specify the scope of their lower bounds and allowed communication protocols. 3. There is not much innovation in the scheme or the lower bound. Both seem to follow the template of [20]. The main contribution is the choice of normalising parameter F_1/C in the scheme which allows the authors to exploit sparsity. I like authors choice of Hoyer's measure of sparsity since it will seamlessly bridge between sparse and non-sparse data and will not require a separate evaluation of sparsity before deciding which scheme to use. But the technical contribution beyond prior work is minimal and doesn't suffice for publication at NeurIPS. *Clarity* I think the presentation is rathe clear. Personally I would have preferred more exhaustive set of experiments with applications in distributed optimisation, such as those in [2], but I am happy that the authors made an effort to compare their results with those in [20] and repeated the same experiments. Also, the reported improvement shows that indeed the data in these experiments comprises sparse observations, a point which was raised in some earlier work as well. Even the SGD updates for training CNNs on MNIST are sparse. [Update after reading the rebuttal] I have to admit that my first reading of the paper was less thorough than I wanted it to be, I will blame the short review cycle for it. I read a few sections again in order to fix the "factual errors" in my review. Here are my additional comments: 1. Lower bound: For some reason, I saw the distribution and information complexity popping-in, and just assumed that there is a Fano bound hidden somewhere. Indeed, this was a gross oversight on my part, apologies. Looks like the main idea was to use Lemma 3.4 to relate MSE to the conditional entropy H(X|\Pi) for a specific distribution. It is an interesting argument, but its relation to existing lower bounds in distributed Gaussian mean estimation has not been clarified. Since the proof doesn't look difficult, I can't comment on the innovation here. 2. The scheme: I think my evaluation of the scheme was not a "major factual error." It is indeed very simple extension of prior work: The main difference is the choice of F_1 to normalize the values so that the clients can use appropriate interval for uniform quantization. But this F_1 had to be computed based on norms of all the vectors and prior work was not at all looking at this case. The authors have clarified how their goal was to capture "local sparsity" and sort of global sparsity at the same time. That's interesting. Overall, I think the paper is acceptable for NeurIPS.

[Author Response · NeurIPS 2019]

**R2**: "The main shortcoming of your algorithm though is from the practical side: you need to synch the normalization factor, which means that your algorithm is two-rounds."

Yes, our algorithm needs an extra round. It is because sparsity can be caused by either dimension unbalance (a few dimensions have much larger values than others) or data unbalance (a few clients have vectors with much larger norms than others). To handle data unbalance and obtain a bound depending on the global sparsity, some form of coordination is necessary, otherwise clients would have no idea whether their vectors are heavy or light compared to others.

However, we would like to point out that, our algorithm can also be run without the synchronization round. For this setting, we can derive a communication bound for each client by simply setting $n = 1$ in Corollary 2.4, although $s$ in the bound will become the local sparsity of the client when doing so. Local sparsity bound is worse than global sparsity when there is data unbalance, but the bound is still better than prior work as long as there is dimension unbalance. This is also verified in the experiments: In Fig 1(a)(c)(d), the data sets used do not have data unbalance (meaning the coordination round is effectively useless), and the results are still better than previous methods.

**R3**: "The lower bound seems to only consider SMPs and doesn't allow for the downlink communication by the centre. In this sense, I feel that the optimality of the proposed scheme has not been established in a strict sense."

*Major factual error:* The lower bound is proved in the broadcast model (see Sec 1.3, as well as Theorem 3.1). The broadcast model allows multi-round protocols and free downlink communication (i.e., each message is public). Thus, lower bounds proved in the this model also hold for other communication models, such as SMP or the message-passing model.

Our upper bound is derived under the message passing model only using private communication. Therefore, the optimality of the proposed scheme has been established in a strong sense: no protocol can beat the proposed scheme (up to a constant factor), even if that protocol is given free downlink or broadcast communications.

**R3**: "The resulting lower bound is obtained by Fano's inequality."

*Factual error:* We have not used or even mentioned Fano's inequality in the paper. Our proof is based on the multiparty information complexity framework, which is different from techniques for proving minimax lower bounds, e.g. Fano's. This is also the reason why our lower bound hold for the broadcast model.

**R3**: "some of which the authors have not cited (such as the ATOMO algorithm which provides a method for handling sparse vectors)"

*Factual error:* We have cited this paper, which is [22] in the reference; and we explained the difference in section 2.

**R3**: "Authors have a rather simple scheme, but it still requires coordination between the parties by the centre."

Please see our response to R2 above.

**R3**: "I feel that random rotations such as randomized Hadamard transform should have been used to avoid this."

(1) Sparsity can be caused by either dimension unbalance (a few dimensions have much larger values than others) and data unbalance (a few clients have vectors with much larger norms than others). Our scheme handles both types of sparsity. Random rotations cannot handle data unbalance, and even worse, it *destroys input sparsity*. (2) Even if there is no input sparsity, the cost of the random rotation method is sub-optimal by a log factor. It is worse than the variable length coding method from the same paper both theoretically and empirically (see [20]), so we didn't say much about it in our paper. (3) The random rotation method needs public randomness, since all clients and the server need to use the same random matrix. This in practice also requires coordination between the parties. See our response to R2 above for more comments on this issue.

**R3**: "There is not much innovation in the scheme or the lower bound. Both seem to follow the template of [20]."

*Major factual error:* Our lower bound does not follow the template of [20] at all. We adopt the information complexity framework and use Yao's minimax principle; the main technical part is a proof of the lemma which roughly says if a combinatorial rectangle contain too much entropy then a random input in it must have large variance. On the other hand, the lower bound in [20] relies on the statistical lower bound from [27]. To get the lower bound, the authors of [27] apply classical techniques for proving minimax lower bounds, which are quite different from ours. We have emphasized in the paper that we cannot simply use the same idea as in [20], because current results on statistical estimation is insufficient to obtain the desired lower bounds. As a result, our lower bound is able to exploits sparsity and is better than [20]. And it holds in the broadcast model as opposed to just independent protocols as in [20].

We agree that our protocol uses a similar framework as prior work (randomized quantization and encoding). However, the main contribution is its sparsity-sensitive analysis and the resulting tight bounds. In addition, the particular quantization and encoding methods are new in this framework, which are important for obtaining the tight bounds.

[Meta-Review · NeurIPS 2019]

This paper provides order-optimal results for distributed mean estimation when the vectors are sparse. With matching upper and lower bounds, the results form a nice and fairly complete story. In terms of techniques, however, the achievable scheme doesn't really require any "new tricks": in a sense the message of the paper is that an appropriately chosen uniform coordinate-wise quantizer is the right thing to do. The reviewers have made several suggestions that could improve the paper, in particular with respect to describing the results and the experiments. A bit more careful accounting of the rounds of communication would avoid confusion. With those changes the paper would be very solid. Since the scheme is not so baroque, I would encourage the authors to make their paper inviting to those outside the area (e.g. practitioners) since simple schemes that provably work are often quite welcome.